# Upscaling of integrated photoelectrochemical water-splitting devices to large areas

Bugra Turan[1], Jan-Philipp Becker[1], Félix Urbain[1], Friedhelm Finger[1], Uwe Rau[1] & Stefan Haas[1]

Photoelectrochemical water splitting promises both sustainable energy generation and energy storage in the form of hydrogen. However, the realization of this vision requires laboratory experiments to be engineered into a large-scale technology. Up to now only few concepts for scalable devices have been proposed or realized. Here we introduce and realize a concept which, by design, is scalable to large areas and is compatible with multiple thin-film photovoltaic technologies. The scalability is achieved by continuous repetition of a base unit created by laser processing. The concept allows for independent optimization of photovoltaic and electrochemical part. We demonstrate a fully integrated, wireless device with stable and bias-free operation for 40 h. Furthermore, the concept is scaled to a device area of $64\,cm^2$ comprising 13 base units exhibiting a solar-to-hydrogen efficiency of 3.9%. The concept and its successful realization may be an important contribution towards the large-scale application of artificial photosynthesis.

[1] IEK5—Photovoltaik, Forschungszentrum Jülich GmbH, 52425 Jülich, Germany. Correspondence and requests for materials should be addressed to B.T. (email: b.turan@fz-juelich.de).

In the last decades, renewable energy sources gained a significant share in the world-wide energy supply. Owing to the fluctuating nature of renewable energy sources, the challenges for further implementation gradually shift from energy generation to energy storage. This trend becomes especially obvious in the fact that research on direct photocatalytic water splitting, which as a research topic dates back to the 1970s (ref. 1), has regained considerable interest during recent years[2–6]. This research progress is driven not only by advancements in materials science concerning new photovoltaic absorber materials[7] and novel catalysts[8–11], but also by design optimizations of multiple-junction solar cells[12–15].

However, in view of the urgent need to develop the technology towards large-scale applications, it appears staggering that research still is almost exclusively focused on laboratory experiments. Comparatively little effort has been devoted to the design and realization of large area or at least scalable devices[16,17]. Taking the recent progress of photovoltaic energy conversion as a paradigm for successful implementation of a renewable energy technology[18–20], it becomes clear that in the end cost effectiveness becomes as much a question of clever device design and process engineering as a question of optimized components. This especially applies for water-splitting devices, which require both a proper management of photons and electrons in the photovoltaic part and of the ions in the electrochemical part. Moreover, many trade-offs between optimizing different components show up only in view of completed, fully integrated devices.

The present paper introduces the design and realization of monolithically integrated solar water-splitting modules based on silicon thin-film module technology. The realized devices fulfill the basic requirements for a future large-scale technology, that is, they are wireless and perfectly scalable to arbitrary device areas. The scalability is achieved by a continuous repetition of a base unit, which in itself combines a photovoltaic (sub) device with the two electrodes of an electrolyser.

Here we take advantage of the laser-patterning processes used for the series connection of thin-film solar cells[21–23] and the wide range of design options going along with this type of processing. Three realized devices are presented. The base unit of the photovoltaic water-splitting device utilizes either a series connection of three a-Si:H single-junction cells or two a-Si:H/μc-Si:H tandem cells connected in series. The first device consists of a singular base unit designed to access the photovoltaic and electrochemical data individually and to evaluate the faradaic efficiency. A second device is built to investigate the operation stability. The third device corresponds to a large-area module (device area $A = 64\,cm^2$) encompassing 13 base units. The latter represents one of the very few practical demonstrations of a scalable monolithic water-splitting concept reported in the literature.

of water oxidation, the overpotential losses at anode and cathode and ohmic losses in the device. For state-of-the-art catalysts, a voltage of ∼1.7–1.8 V is needed for a current density of $10\,mA\,cm^{-2}$ at the electrodes[8]. Such high voltages can be generated by single-junction solar cells with high band gap energy[24] ($\sim E_g \gtrsim 2.2\,eV$) or by the use of multiple junctions connected in series[14]. Since the optimal band gap for single-junction solar cells under AM1.5G illumination is 1.34 eV, the use of high band gap solar cells leads to high optical transmission losses and thus to inefficient solar cells[25,26]. In contrast, the utilization of multijunction solar cells, spatially neighbouring interconnected solar cells, or a combination of both offers more freedom with respect to the optimization of the device[27]. In the present work, both types of series connection approaches were used.

As shown in Fig. 1, all laboratory experiments may be categorized into three types, dependent on the number of wires required. In addition, Fig. 1 illustrates how the three types can be converted into a scalable module design. The upscaling is achieved by a continuous repetition of one base unit as indicated by the dashed boxes in Fig. 1d–f. Figure 1a depicts an approach

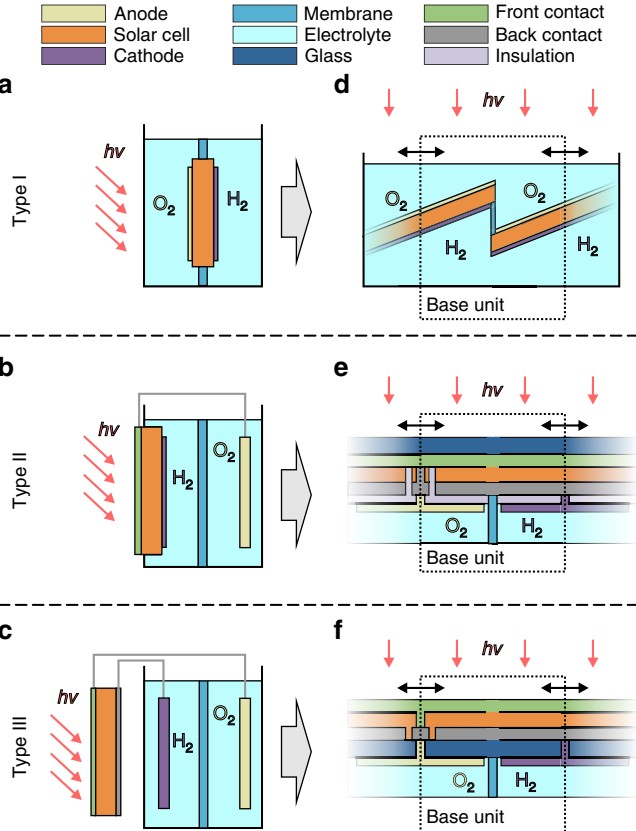

**Figure 1 | Classification of different photovoltaic water-splitting designs.** The set-ups typically used in laboratory experiments are shown in **a–c**, while corresponding designs of possible large devices are shown in **d–f**. For the laboratory experiment (**a**) and the upscaled design (**d**) the PV part is completely immersed in the electrolyte with the catalyst electrodes on both sides (type I). For type II (**b,e**), only one side of the PV part is exposed to the electrolyte. Note that a wire connection between the PV front contact of the PV element and the back electrode of the EC is needed for the laboratory experiment (**b**), whereas the monolithic design of the scalable device is wireless (**e**). Type III represents an approach with completely separated PV and EC elements. In the laboratory set-up (**c**) two wire connections are needed. The corresponding monolithic (**f**) approach needs electrical contacts through the substrate but is, again, wireless.

## Results

**General design considerations.** Any form of a solar photoelectrochemical water-splitting device comprises a series connection of a photovoltaic cell (PV) that converts solar photons in electrons and holes, and an electrochemical cell (EC) with two half-reactions: $2H_2O + 2e^- \rightarrow H_2 + 2OH^-$, the hydrogen evolution reaction and $4OH^- + 4h^+ \rightarrow O_2 + 2H_2O$, the oxygen evolution reaction (in an alkaline electrolyte). The electrical circuit is closed via transport of $OH^-$ ions in the electrolyte, whereas $H_2$ and $O_2$ are evolving at the cathode and the anode, respectively. Finally, a membrane or a glass frit prevents mixing of the two gases.

Water splitting only takes place if the solar-generated electron and holes have enough energy to overcome the energetic barrier

where the whole device is immersed in the electrolyte (type I). Examples for this design were reported by Lin et al.[28], Reece et al.[29] and recently, including an anion-exchange membrane, by Verlage et al.[30]. A scalable adaption via the 'louvered cell' (cf. Fig. 1d) design has been introduced by Walczak et al.[17] using Si and $WO_3$ as photoabsorbers. This device represents one of the rare practical examples of a water-splitting module consisting of more than one base units (two units in the demonstrated design). So far, a solar-to-hydrogen (STH) efficiency of only $\eta_{STH} = 0.24\%$ has been demonstrated for an operation time of more than 24 h. Figure 1b shows a design with one wire (type II). This approach is often used for systems embracing superstrate thin-film solar cells (illumination through the glass support)[31–33]. This has the advantage that optical and electrochemical properties can be optimized independently because of the spatial separation. Solar cells based on superstrate technology can be transformed to a scalable water-splitting device by the approach depicted in Fig. 1e with a coplanar electrode configuration. A realization of a base unit employing a-Si:H/a-Si:H tandem cells has been demonstrated as early as 1985 by Appleby et al.[34]. They reported a STH efficiency of $\eta_{STH} = 2.6\%$. The series connection of the two tandem cells in the unit was achieved by wiring. Yamada et al.[35] reported a wireless base unit using a thin-film silicon triple-junction solar cell. A type II concept was also used by Verlage et al.[30] integrating a GaAs/InGaP tandem cell into a self-standing water-splitting device of an active area of 1 cm². 

Finally, it is possible to keep the photovoltaic unit completely outside of the electrolyte (Fig. 1c, type III) using two wires. On the laboratory scale, this approach is used to illustrate the compatibility of solar cell technologies with an EC of the same area[36–38]. As shown in Fig. 1f an upscaling of this configuration is also feasible. Recently, Jacobsson et al.[39] have demonstrated an intermediate step to a scalable substrate device based on the series connection of three $Cu(In,Ga)Se_2$ solar cells, which exhibits an efficiency of $\eta_{STH} = 10\%$. However, in this device metal stripes were used for the electrical connection to the water-splitting electrodes. For an arbitrarily scalable device, the electrical connections could be realized by holes through the substrate as illustrated in Fig. 1f.

Each of the introduced upscaling concepts exhibits inherent advantages and disadvantages. Apparent disadvantages of type I (Fig. 1d) are the inevitable illumination through the electrolyte[40] and the complex mechanical construction. These drawbacks are avoided by types II and III. However, for type II an additional electrical insulation is required and for type III contact formation through the substrate can be challenging.

For the practical demonstration of a scalable, wireless and monolithic solar water-splitting device, we chose type II because

this concept can be realized using mature technologies for large-scale production. Furthermore, scarce materials were avoided.

**Device realization.** For the present work we realized solar water-splitting modules following the design concept II shown in Fig. 1e. The devices were either based on three series-connected single-junction a-Si:H solar cells or two series-connected a-Si:H/ μc-Si:H tandem cells to generate the required voltage to sustain electrolysis (cf. Supplementary Figs 1 and 2 for an equivalent circuit diagram). We chose these different types of absorber configurations to illustrate the flexibility of the design. However, the design can also be used with high band gap absorbers or stacked solar cells with higher voltages to generate the needed voltage without a lateral side-by-side series connection. Figure 2 shows a cross-sectional sketch of the proposed concept for the case of three spatially neighbouring series-connected solar cells used as a base unit. The PV element was designed in superstrate configuration, which is often used in thin-film silicon, CdTe (ref. 41), dye-sensitized[42] or organometal halide perovskite[43] solar cells.

The series connection of the individual solar cells was realized by the introduction of a structure that connects the back contact of one cell with the front contact of the adjacent cell. This structure was created by selective laser ablation in between the layer-deposition steps.

As apparent in Fig. 2, the anodes and cathodes were placed side-by-side on the back side of the PV element. Note that neighbouring base units share electrodes. Hence, the sequence of the interconnection needed to be alternating. Thereby, the design avoids additional active area losses because of further laser scribes.

The insulating epoxy layer was used as a corrosion protection against the alkaline electrolyte as well as electrical insulation. Chemically resistant epoxy resins have proven to be stable when 1 M KOH was used as electrolyte. In addition, a conductive polymer resin was applied at the electrical interfaces between PV element and the anodes/cathodes. Nickel-filled epoxy fulfilled the requirements for both chemical protection and electrical conductivity.

For the sake of simplicity, bare nickel foam was used for both anode and cathode. Furthermore, for this proof of concept no membrane was employed. However, preliminary results demonstrating gas separation in a single base unit using an anion exchange membrane are shown in the Supplementary Fig. 3.

The PV front end and the EC back end can be optimized independently because they are merely coupled by the insulating

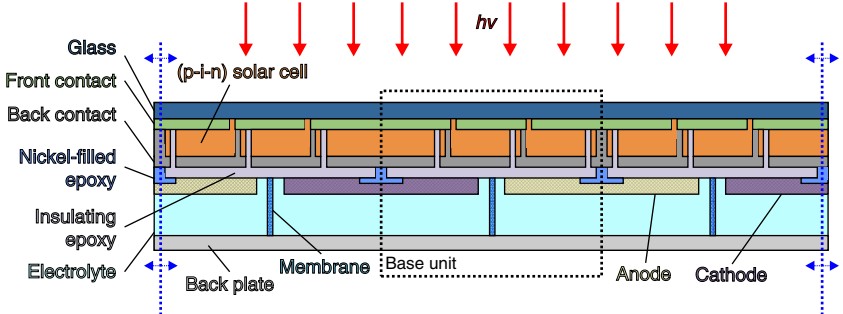

**Figure 2 | Device cross-section.** The sketch shows the device structure of a scalable, fully integrated photovoltaic water-splitting device in the superstrate configuration. The number of cell stripes in series can be easily adjusted (three in this case). Dimensions are not to scale in width and thickness of the layers. The sketch only shows an excerpt of the module. The configuration can be extended in both directions (hinted by the dashed blue arrows). The base unit that defines the region of periodic repetition is depicted by the dashed box.

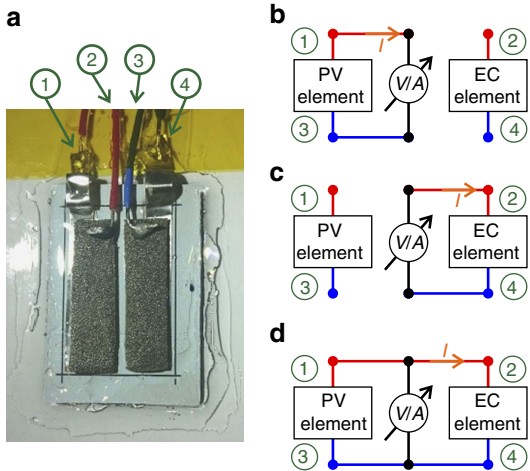

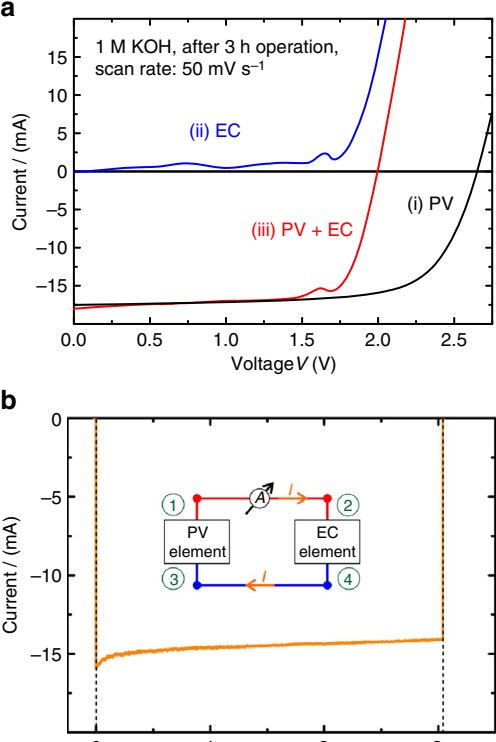

**Figure 3 | Device for faradaic efficiency assessment.** Shown in **a** is a photograph of device #1 with separated, electrically insulated, contacts for characterization of the individual properties of PV and EC elements. A series connection of three cell stripes, each with a length of 27 mm and a width of 5 mm, leads to a device area of 4.8 cm². Shown in **b**–**d** is a block diagram of the device and different operating modes, (**b**) for measuring the photovoltaic part, (**c**) for measuring the electrochemical part and (**d**) for the combined measurement. The numbers in green depict the device terminals.

and conductive polymer coatings. Thus, the concept allows for the utilization of various PV and EC technologies.

**Assessment of the faradaic efficiency.** In a first step the presented concept was realized by processing of one base unit (device #1) with the design from Fig. 2 (dashed box). A module with three a-Si:H solar cells connected in series was used here.

For the evaluation of the faradaic efficiency, the correlation of the gas generation rate to the current through the system is required. However, the wireless design of the device merely allows monitoring of the operating voltage point of the system. Therefore, a special monitoring module was built that decouples PV and EC elements. This was achieved by the use of an insulating epoxy layer instead of the nickel-filled epoxy depicted in Fig. 2. Wires were soldered to the contacts of the PV and EC elements. Thereby, the current can be monitored as well. Figure 3 shows a photograph of such a module and a block diagram illustrating the different measurement modes.

As shown in Fig. 3b–d three operating modes of the device are possible: (b) PV operation, (c) EC operation and (d) simultaneous monitoring of the current and voltage during water splitting.

Eventually, this device was used to determine the faradaic efficiency of the system by a correlation between the experimental gas rate and the theoretical gas rate calculated from the current during operation. The graphs in Fig. 4a depict the current–voltage (I–V) characteristics of this monitoring module after 3 h of operation.

From Fig. 4a it can be seen that in each operating mode the individual subdevice characteristics can be evaluated. The superposition of the individual characteristics in operating modes (i) and (ii) match the I–V curve in the combined PV + EC operating mode (iii).

The current through the system during photovoltaic water splitting is plotted in Fig. 4b over the operating time. The measurement mode is illustrated by the inset. With Faraday's law of electrolysis and the ideal gas law the theoretical evolution rate of oxygen and hydrogen can be calculated from the current with

**Figure 4 | Individual device characteristics.** The I–V characteristics of device #1 in different operating modes. (**a**) Individual and combined I–V curves of the PV and EC elements after 3 h of operation under illumination. The I–V sweeps were all performed starting from V = 0 V. (**b**) Operating current of device #1 under illumination as a function of time with the operating mode depicted by the inset.

the following relation:

$$r = \frac{\mathrm{d}V}{\mathrm{d}t}\frac{1}{A} = \eta_F \frac{RT}{F \times p \times z \times A} I \tag{1}$$

The gas rate $r$ is related to the current $I$ multiplied with a factor consisting of the ideal gas constant $R$, the temperature $T$, Faraday's constant $F$, gas pressure $p$ and the ratio between charge transfer and gas molecules formed $z$, which is 1.33 for water splitting. For the sake of comparability the gas rate was normalized by the illuminated device area $A$. Furthermore, the factor $\eta_F$ can be regarded as the faradaic efficiency, which is ideally unity.

The theoretical gas rate from the current measurement can be compared with the gas rate that was measured by collecting the co-evolved reaction products within a bell jar. For simplicity reasons, a membrane was not used in this experiment and hydrogen and oxygen were collected together. Figure 5 shows both values as a function of time.

It can be seen that the calculated and the experimental values merge after a certain time delay, which is an indication that the faradaic efficiency of the device was in fact close to unity.

**Device operation stability.** For the investigation of the device durability, a fully integrated photovoltaic water-splitting device was prepared with nickel-filled epoxy as an electrical connection between PV and EC elements, as intended in the design in Fig. 2. A photograph of this device (referred to as device #2) glued on a glass substrate for clamping into the measurement set-up is shown in Fig. 6a.

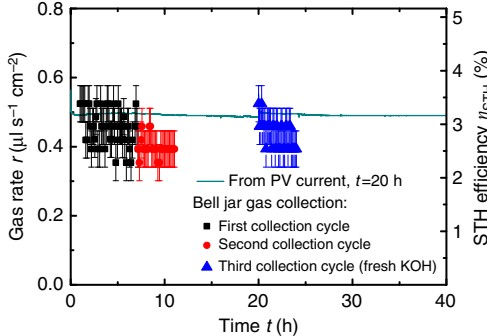

**Figure 5 | Gas flow rate comparison.** The graph shows the gas flow rate $r$ of device #1 as a function of time for a theoretical calculated gas rate from the current measurement (black curve) and gas rate evaluated from collection of reaction products with a bell jar (red circles). The systematic error originates from the evaluation of volume scale bar and time stepping and is assessed with $0.06\,\mu l\,s^{-1}\,cm^{-2}$.

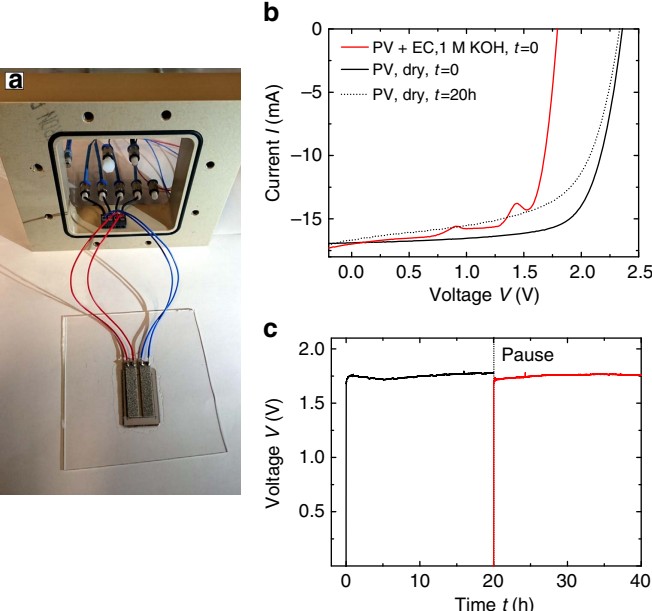

**Figure 6 | Photograph and measurements of a single base unit.** (**a**) Photograph of a device consisting of one base unit with three a-Si:H solar cells connected in series (device #2). Both anode and cathode are placed on the back side and consist of nickel foam. For the measurement of the electrical properties, a four-point connection is soldered to the EC electrodes. A series connection of three cell stripes, each with a length of 29.8 mm and width of 5 mm, leads to a device area of 5.3 cm². The device was clamped into the gas-tight fixture made of polyether ether ketone (PEEK). The graph in **b** shows the $I$–$V$ characteristics of device #2 during illumination in 1 M KOH as well as the PV element's characteristics before and after 20 h of operation. The graph in **c** shows the operating voltage of the device during illumination in 1 M KOH as a function of the time.

For the characterization of the electrical properties, two wires were soldered to each EC electrode to avoid any influence of the contact resistances. Thus, only the parallel connection of both PV and EC elements can be evaluated. During photovoltaic water splitting, it is only possible to monitor the operating voltage of the device. However, without the electrolyte it is possible to characterize the PV element.

**Figure 7 | Gas rate evaluation.** The graph shows the measured gas rate of device #2 with three series-connected a-Si:H solar cells and nickel-foam electrodes in 1 M KOH under illumination. After 20 h the electrolyte was removed for the characterization of the device and the experiment was resumed in a fresh electrolyte. The gas rate was evaluated from three gas collection cycles as indicated by the different symbols. The solid line indicates the theoretical gas rate calculated via equation 1 using the current extracted from the operating voltage of the PV element's $I$–$V$ curve after 20 h of operation (see Fig. 6b). The STH efficiency, as calculated with equation 2, is shown on the right hand $y$ axis. The systematic error originates from the evaluation of volume scale bar and time stepping and is assessed with $0.06\,\mu l\,s^{-1}\,cm^{-2}$.

Figure 6b shows the $I$–$V$ characteristics of the device under illumination in 1 M KOH (PV + EC) as well as without the electrolyte (PV) before and after 20 h of operation.

A degradation of the photovoltaic $I$–$V$ characteristics was observed after 20 h, mostly because of a reduction of the fill factor. This can be attributed to the well-known Staebler–Wronksi effect for amorphous silicon technology[44]. The degradation rate is initially stronger before it saturates for higher operation times[45]. The electrolyte was removed for the characterization after 20 h and the experiment was resumed with a fresh 1 M KOH solution.

The $I$–$V$ curve of the whole system (PV + EC) shows an operating voltage ($I = 0$ mA) of ~1.75 V. The nickel foam exhibits a very large surface-to-volume ratio of ~6,900 m² m⁻³. The large surface area leads to a low current density and, consequently, results in a low overpotential. For an optimization of the device, the PV element's properties need to be adjusted to fit the EC characteristics and *vice versa*. The integrated series connection provides an additional degree of freedom for the adjustment of the $I$–$V$ characteristics of the PV element.

In Fig. 6c the operating voltage is monitored during water splitting over the accumulated period of 40 h. The constant voltage indicated an excellent stability of the device during this period. After 20 h the experiment was halted for characterization and resumed afterwards for an additional 20 h.

The co-evolved gaseous reaction products were collected. Figure 7 shows the evaluated gas rate as a function of time.

The different symbols correspond to three gas volume collection sequences. After each sequence the gas was removed from the bell jar. The evaluated gas rate was in the range of $0.35$–$0.55\,\mu l\,s^{-1}\,cm^{-2}$ during the experiment. A certain reduction is observed as seen in the second evaluation (*cf.* circles in Fig. 7). However, to some degree the initial gas rate was recovered after exchange of the electrolyte (*cf.* triangles in Fig. 7). Supplementary Movie 1 shows a time-lapse photography of a similar device in operation for an extended period of time.

The current through the system cannot be extracted because of the structure of the device. However, with the characterization of the $I$–$V$ curve of the PV element it is possible to correlate the

operating voltage to the equivalent operating current. The line curve in Fig. 7 indicates the theoretical gas rate calculated from this current via Equation 1. For the complete time period of 40 h the PV element's $I$–$V$ curve of the device after 20 h was used for current-operating point extraction.

It can be seen that the calculated gas rate correlates well with the experimental gas rate evaluation. However, there is a slight overestimation of the calculated gas rate. This discrepancy may be explained by a time-dependent change of the PV element's $I$–$V$ curve. For instance, the system temperature that differs before, during and after device operation (Supplementary Figs 4–6 indicate the influence of the operation on the device). For a reliable assessment of the device performance, a quantification of the evolved gases is favourable.

The gas rate of the system can be used to evaluate the STH efficiency of the device.

$$\eta_{STH} = \frac{2}{3} \frac{r \times \Delta G}{P_{IN} \times V_m} \qquad (2)$$

In Equation 2, $r$ is the gas rate, the factor 2/3 corresponds to the hydrogen-to-oxygen ratio, $\Delta G = 273\,kJ\,mol^{-1}$ is the change in Gibb's free energy per mole of $H_2$, $V_m = 24.81\,mol^{-1}$ is the molar volume of an ideal gas and $P_{IN} = 100\,mW\,cm^{-2}$ is the illumination power density. From equation 2 follows that a STH efficiency of 1% corresponds to a gas rate of $r = 0.155\,\mu l\,s^{-1}\,cm^{-2}$.

Furthermore, the STH efficiency can be calculated from the current $I$ through the system[46].

$$\eta_{STH} = \frac{1.23V \times I \times \eta_F}{P_{IN} \times A} \qquad (3)$$

In equation 3, $\eta_F$ is the faradaic efficiency and $A$ is the device area. The knowledge of the spectrum and the intensity of the used solar simulator are crucial for a correct determination of the STH efficiency[47].

From Fig. 7a the mean value of the device area STH efficiency of $\sim 3\%$ was observed for device #2 for more than 20 h. This corresponds to an active area STH efficiency of 3.3% (see Supplementary Fig. 7). A better matching of the PV and the EC element's characteristics could increase the STH efficiency. Furthermore, the electrochemical activity of the nickel foam can be increased significantly by a modification with suitable catalysts[48–50]. The utilized a-Si:H single-junction solar cells could ideally yield a STH efficiency of 5.4% (calculated from data in Supplementary Fig. 8).

To estimate the potential of our concept that enables more possibilities with regard to the $I$–$V$ output characteristics, a simple calculation was made for various PV technologies with the best reported catalysts. Table 1 shows the theoretical STH efficiencies achievable with reported values from the literature.

Depending on the PV technology the number of series-connected cells was varied to achieve matching to the EC

element's required voltage of 1.7 V (NiMo and NiCo catalysts and no additional losses). The output voltage of the solar cell was multiplied by the number of cell stripes, while the current density was divided accordingly. Interconnection losses were neglected.

It should be emphasized that most of the shown PV technologies cannot sustain the required voltage of 1.7 V without a series connection. With our design concept, large area, scalable, efficient photovoltaic water-splitting devices are feasible for all thin-film PV technologies where a series connection can be realized, for example, by laser processing[51–54].

**Upscaling to large-area devices**. In the previous section, the successful realization of one base unit of our concept was demonstrated and it showed stable operation over 40 h. Next, the upscaling of the concept with multiple adjoining base units will be presented (device #3). In contrast to the previous experiments (device #1 and #2), each base unit consisted of two series-connected a-Si:H/μc-Si:H tandem junction solar cells. This allows for closer packing of the base units on the substrate. In total, 13 adjoining base units were prepared on a $10 \times 10\,cm^2$ substrate. The overall area of the device was $A = 64\,cm^2$. Figure 8 shows a photograph of this device after preparation.

The cell stripe width for each base unit was 2.5 mm, which is not ideal in terms of solar module efficiency[55] but rather is chosen to illustrate the successful upscaling of the concept by having a multitude of adjoining base units. The cell stripe length was 80 mm and the interconnection width was 300 μm. Since the insulating epoxy was deposited manually, a rather wide fillet to the front contact of 2 mm was required, which decreases the active PV area. However, with an optimized laser-patterning process a fillet width as narrow as 100 μm is possible, which increases the active area[56,57].

For this fully integrated device #3, monitoring of the voltages is not feasible and because of the symmetric design of the interconnection the base unit's PV element properties are not accessible. Therefore, proper upscaling behaviour was evaluated by the measurement of the gas rate. Figure 9 shows the overall collected gas volume as well as the gas rate as a function of the operating time.

Owing to the high overall gas rate the bell jar needed frequent clearing to reset the collection as indicated by the different symbols in Fig. 9. A decrease in the gas rate over time was observed. However, after a pause overnight (marked with a dashed line in Fig. 9) of the experiment, the initially higher gas rate was partially recovered, which indicates a variation of the operating condition over time rather than a degradation of the device.

The successful upscaling of the concept becomes apparent when the gas rates of device #2 ($r \approx 0.4\,\mu l\,s^{-1}\,cm^{-2}$, see Fig. 7) and device #3 ($r \approx 0.6\,\mu l\,s^{-1}\,cm^{-2}$, see Fig. 9) are compared. The device area was increased from 5.3 to 64 $cm^2$ while the gas rate

**Table 1 | Calculated maximal solar-to-hydrogen efficiencies.**

| PV technology | Current density divided by number of cell stripes (mA cm$^{-2}$) | Number of solar cell stripes | Potential solar-to-hydrogen efficiency (%) | Reference |
|---|---|---|---|---|
| Silicon triple junction | 7.7 | 1 | 9.5 | Urbain et al.[14] |
| Perovskite solar cell | 11.7 | 2 | 14.4 | Yang et al.[43] |
| Cadmium-telluride (CdTe) | 10.1 | 3 | 12.4 | Green et al.[67] |
| Cu(In,Ga)Se$_2$ (CIGS) | 11.5 | 3 | 14.2 | Green et al.[67] |

EC, electrochemical; PV, photovoltaic.
State-of-the-art PV technology is used, while for the EC element NiMo and NiCo were used as catalysts with their respective overpotentials at 10 mA cm$^{-2}$ from McCrory et al.[8]. Any $IR$-drop in the electrolyte and additional losses were neglected, which leads to an overall water-splitting voltage of 1.7 V. The current density at 1.7 V was divided by the number of cell stripes.

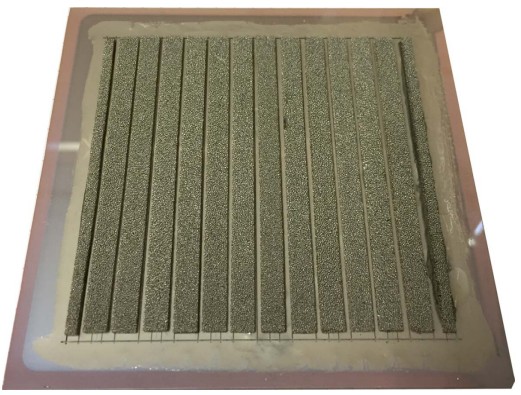

**Figure 8 | Photograph of a large-scale photovoltaic water-splitting device.** The total device area was 64 cm² with an active area of 52.8 cm². Each base unit consists of two series-connected a-Si:H/μc-Si:H tandem solar cells with a cell stripe width and length of 2.5 and 80 mm, respectively. Thirteen base units were neighbouring on a 10 × 10 cm² substrate. The back end was made of laser-cut nickel-foam elements for both cathodes and anodes. A photograph from the front side of a similar device can be found in Supplementary Fig. 9.

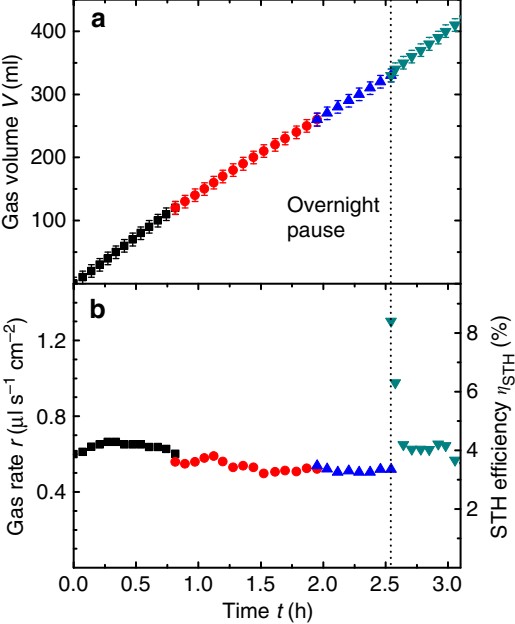

**Figure 9 | Large-area device gas rate evaluation.** The graphs show the collected gas volume (**a**) and gas rate/STH efficiency (**b**) of an upscaled water-splitting device (device #3) as a function of time. The total device area was 64 cm² with 13 neighbouring base units. Each base unit consisted of two series-connected a-Si:H/μc-Si:H tandem solar cells. The cell stripe width was 2.5 mm with a length of 80 mm. The individual symbols/colours in the graphs correspond to gas volume collection sequences after which the collection bell jar needed to be emptied. The dashed line marks where the experiment was paused overnight. The systematic error originates from the evaluation of volume scale bar and time stepping and is assessed with 10 ml and 0.06 μl s⁻¹ cm⁻², respectively.

was maintained. The slightly higher gas rate of the upscaled module can be attributed to the better matching of the I–V characteristics of PV and EC elements in the case of the series connection of two tandem solar cells. Such base unit usually exhibits a higher current density compared with three series-connected a-Si:H single-junction solar cells. The extracted device

area (or active area) STH efficiency for device #3 was ∼3.9% (4.7%), which is remarkable for non-optimized device geometries in terms of cell stripe width and catalyst dimensions. The utilized a-Si:H/μc-Si:H tandem solar cells could ideally yield a STH efficiency of 6.6% (calculated from the data in Supplementary Fig. 10) assuming an operation voltage of 1.7 V.

## Discussion

In this study we introduced a thin-film concept for the upscaling of artificial photosynthesis to large areas. The concept is modelled according to the idea of thin-film photovoltaic modules and their integrated series interconnection of single-cell stripes. Analogously, we used water-splitting base units that can be repeated as often as desired to compose a large-area water-splitting module. Note that each base unit acts as an independent water-splitting device. Upscaling by repetition of base units does not introduce additional losses. The realization was demonstrated using thin-film silicon photovoltaic technology and non-precious metal catalysts in alkaline solution. However, the concept is compatible with any thin-film photovoltaic technology.

The proof of concept was presented first, with a single base unit that was modified to access all relevant photovoltaic and electrochemical properties individually. Afterwards, a second single base unit was created, which exhibited excellent stability for more than 40 h of operation under illumination in 1 M KOH. Finally, an upscaled device with an area of 64 cm² and multiple base units has proven the successful scalability of the concept by a similar gas rate per unit area between the upscaled and single base unit device. To our knowledge, this is the first reported scalable and wireless photovoltaic water-splitting device on this scale. For the upscaled device a STH efficiency of ∼3.9% was measured. The next immediate step is the implementation of a gas separation membrane for the upscaled design (cf. Fig. 2). Preliminary experiments on single base unit devices showed a negligible effect of the membrane on the device performance (see Supplementary Fig. 3 for more details).

We believe that these results may encourage the successful application of photoelectrochemical water splitting to system scales that are relevant for a combined renewable energy generation and storage, which is gaining substantial importance in the future.

For a successful application of the concept in a commercial system or product, the final metric of viability is the cost of hydrogen. Several studies addressed the costs for solar hydrogen technologies[58–60]. One common conclusion is that the efficiency is the most sensitive parameter to lower the cost of hydrogen. Therefore, based on the proof of concept, future lines of work need to address efficiency improvements and technoeconomical questions to assess its cost competiveness. A similar procedure was taken for the 'louvered cell' design, which was first introduced and realized[17] and subsequently evaluated with regards to its energy balance[61] and cost competitiveness[62].

Beyond the generation of hydrogen, artificial photosynthesis also addresses the reduction of CO₂ (ref. 63) or the creation of a sustainable closed carbon cycle (for example, solar fuels). In this broader context, many chemical challenges are involved[4]. One key challenge is the supply of a sufficient voltage to sustain the electrochemical reactions. This aspect is successfully addressed by the proposed concepts.

## Methods

**Solar cell deposition.** For all experiments in this work, commercially available 1.1 mm-thick glass substrates coated with fluorine-doped tin dioxide (SnO₂:F) as transparent conductive oxide from the Asahi Glass Company (type U) were used as front contact material[64]. We applied state-of-the-art thin-film silicon technology for the deposition of p-i-n single-junction solar cells from hydrogenated

amorphous silicon (a-Si:H) and multijunction p-i-n/p-i-n solar cells from hydrogenated amorphous and microcrystalline silicon (a-Si:H/μc-Si:H) using plasma enhanced chemical vapour deposition[65]. A layer stack consisting of aluminium-doped zinc oxide (ZnO:Al) and silver (Ag) was deposited as back contact material by radio frequency (RF) magnetron sputtering[66].

**Device assembly.** For corrosion protection and electrical insulation of the system's back end, a chemically resistant epoxy from Loctite (type 9483) has proven to be a suitable two-component adhesive for manual patterning of the PV element's back side. The contact to the PV element was masked by an adhesive tape, which was removed after homogenous application of the epoxy and before annealing on a hot plate for 20 min at a temperature of 100 °C. Subsequently, a one-component nickel-filled epoxy from Alfa Adhesives (type E10-102) was applied on the exposed contacts of the PV element. On top of this adhesive conductive epoxy, 1.4 mm-thick nickel-foam electrodes (RECEMAT BV, Ni-5763) were attached and the whole device was annealed for 120 min at 125 °C. The individual nickel-foam stripes were laser-cut to the required width and length by a CO$_2$ laser tool.

**Measurement set-up.** Single base units were glued with epoxy (Loctite 9483) on a $10 \times 10$ cm$^2$ glass substrate and clamped into a hermetically sealed sample holder made from polyether ether ketone. Four wires were fed into the chamber as well as tubing for electrolyte filling and extraction of the gaseous reaction products during the water-splitting experiments. For the upscaled water-splitting devices on $10 \times 10$ cm$^2$ substrates, an additional glass substrate for mechanical support was not required. Single base units were illuminated with a small-area class ABB solar simulator from Newport (type LCS-100 model 94011A). The distance between light source and device was adjusted to generate an incident illumination intensity of 1 kW m$^{-2}$ with the help of reference measurements using a spectrophotometer together with calibrated solar cells. The spectrum of the sun simulator was measured simultaneously with the reference. For large-scale devices, an in-house built large-area sun simulator was used. All spectra can be found in Supplementary Fig. 11. In addition, various photographs of both measurement set-ups are shown in Supplementary Figs 12 and 13.

**Electrochemical characterization.** All experiments were conducted in an aqueous 1 M potassium hydroxide (KOH) solution prepared from analytical-grade chemicals (Merck). The deionized water (Millipore) used for the preparation of the solution was scrubbed with air for 15 mins for the saturation with oxygen. To illustrate the influence of the concentration on the system, a comparison of the EC element's $I$–$V$ characteristics in 0.1 and 1 M KOH is shown in Supplementary Fig. 14. A potentiostat/galvanostat from Gamry Instruments (Reference 600 + ) was used for the electrochemical characterization as well as for voltage monitoring over time and $I$–$V$ characteristics measurements of the PV element. The potentiostat/galvanostat was either used in a two-wire or four-wire sense mode. If not specifically stated otherwise, a scan rate of 50 mV s$^{-1}$ was used for all measurements with an open-circuit delay of 30 s. For the monitoring of the time-resolved current of device #1 during illuminated operation a multimeter from Keithley (model 2000) was used.

**Measurement of the co-evolved gases.** Hydrogen and oxygen were collected with a bell jar or inverted burette with two different maximum volumes of 100 and 1,000 ml. A camera was focused on the bell jar and took time-lapse photography so that the generated gas volume could be evaluated visually together with the time stamp for prolonged operation durations. Each bell jar exhibits a scale stepping that was 1 and 10 ml, respectively. We estimate the reading error to be ± 1 and ± 5 ml. Before the measurement the deionized water that is used as a gas barrier in the gas collection system was scrubbed with air for 15 min. Supplementary Fig. 13 shows a photograph of the gas collection apparatus.

**Laser material processing.** For the integrated series connection[22] Nd:YVO$_4$ Q-switched diode-pumped solid-state laser sources from ROFIN (type PowerLine E) were used, which exhibit a pulse duration between 7 and 12 ns (full-width at half-maximum). For the front contact insulation process (P1) the third harmonic with a wavelength of 355 nm was used. A pulse repetition frequency of 15 kHz was applied and the average power of the laser was 210 mW. Focusing was performed with an f-theta lens ($f = 108$ mm), which lead to a beam spot radius of 19 μm. A laser source with the second harmonic and a wavelength of 532 nm was used for the removal of the absorber (P2) as well as for the back contact removal process (P3) (refer to Supplementary Methods for details on P1 to P3). The pulse frequencies were either 17 and 11.5 kHz, respectively. For P2 the average power 430 mW, while 290 mW was used for P3. For both processes a lens system with a focal length of $f = 300$ mm was used to focus the laser beam, which led to a beam spot radius of 60 μm. The three processes P1–P3 required an overall interconnection width of 300 μm. All processing were carried out through the glass side. The additional processes required for the short-circuiting between front and back contacts were modified P2 and P3 processes. Details on laser processing can be found in Supplementary Fig. 15.

**Data availability.** The data supporting our findings of this study are available in the article and the Supplementary Information file. All other relevant data are available upon request from the authors.

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

## Acknowledgements

We thank H. Siekmann, A. Bauer, G. Schöpe, J. Kirchhoff and C. Zahren for their contributions to this work. We also thank Tobias Dyck for the photographs of the devices. The research was partly financially supported by the Deutsche Forschungsgemeinschaft (DFG) Priority Program 1613 (SPP 1613) and by the German Bundesministerium für Bildung und Forschung (BMBF) in the network project Sustainable Hydrogen (FKZ 03X3581B).

## Author contributions

B.T. and J.-P.B. designed and performed the experiments. B.T., J.-P.B and F.U. interpreted data. U.R., F.F. and S.H. developed the conceptual idea and supervised the work. B.T., J.-P.B., U.R. and S.H. wrote the manuscript. All authors participated in discussions and contributed to editing of the manuscript.

## Additional information

**Competing financial interests:** The authors declare no competing financial interests.

