## [Peer Review File · Nature Communications]

REVIEWERS' COMMENTS:

Reviewer #1 (Remarks to the Author):

The answers that the authors have given are okay and I would agree that "...the field is in need for a push from the engineering side....", however, that directly implies something pointed towards a commercial product. In this case then, the metric is the projected cost of hydrogen, not the cleverness of the engineering design. It doesn't make a lot of sense to me to publish a multitude of engineering designs if they have no future in producing cost-competitive hydrogen.

There have been a number of technoeconomic studies that can be used to inform and evaluate various engineering designs, these include:

B. D. James, G. N. Baum, J. Perez, K. N. Baum and O. V. Square, Technoeconomic Analysis of Photoelectrochemical (PEC) Hydrogen Production, 2009, https://www1.eere.energy.gov/hydrogenandfuelcells/pdfs/pec_technoeconomic_analysis.pdf.

B. A. Pinaud, J. D. Benck, L. C. Seitz, A. J. Forman, Z. Chen, T. G. Deutsch, B. D. James, K. N. Baum, G. N. Baum, S. Ardo, H. Wang, E. Miller and T. F. Jaramillo, Energy Environ. Sci., 2013, 6, 1983-2002.

P. Zhai, S. Haussener, J. Ager, R. Sathre, K. Walczak, J. Greenblatt and T. McKone, Energy Environ. Sci., 2013, 6, 2380-2389.

Developing an engineering design for a system that is highly unlikely to produce cost-competitive hydrogen does not in my mind lend itself to publication in a high impact journal. Rather it belongs in an engineering journal.

In summary, for these new engineering designs, I'm inclined to request a technoeconomic analysis on them to see what the possibilities are for producing cost-competitive hydrogen. It is more that an engineering design these days, the metric is the final cost of hydrogen and tools are available to put forth a projected cost based on the specific design.

Publication in Nature Communication is not recommended.

Reviewer #3 (Remarks to the Author):

I have now read the new version of the manuscript by Turan. et al. My general impression is fairly unchanged from the last time I saw it. They do construct working devices, the discussion is sound, the presentation is decent and the topic of upscaling and the associated problems are important for the application of water splitting systems and is something that deserves more attention. With the understanding that it is more of an engineering paper I think this manuscript deserves to be published. I think that the changes they have made in response to the reviews comments are reasonable. Think that the comment from rewire 1 concerning the separation of the reaction gases are a bit unfair as incorporating membranes that can do this is a rather straight forward geometrical exercise given the suggested experimental setup.

I have no additional comments and are in favour of acceptance for publication.

REVIEWERS' COMMENTS (08.07.2016):

Reviewer #1 (Remarks to the Author):

The answers that the authors have given are okay and I would agree that "...the field is in need for a push from the engineering side...", however, that directly implies something pointed towards a commercial product. In this case then, the metric is the projected cost of hydrogen, not the cleverness of the engineering design. It doesn't make a lot of sense to me to publish a multitude of engineering designs if they have no future in producing cost-competitive hydrogen.

There have been a number of technoeconomic studies that can be used to inform and evaluate various engineering designs, these include:

B. D. James, G. N. Baum, J. Perez, K. N. Baum and O. V. Square, Technoeconomic Analysis of Photoelectrochemical (PEC) Hydrogen Production, 2009, https://www1.eere.energy.gov/hydrogenandfuelcells/pdfs/pec_technoeconomic_analysis.pdf.

B. A. Pinaud, J. D. Benck, L. C. Seitz, A. J. Forman, Z. Chen, T. G. Deutsch, B. D. James, K. N. Baum, G. N. Baum, S. Ardo, H. Wang, E. Miller and T. F. Jaramillo, Energy Environ. Sci., 2013, 6, 1983-2002.

P. Zhai, S. Haussener, J. Ager, R. Sathre, K. Walczak, J. Greenblatt and T. McKone, Energy Environ. Sci., 2013, 6, 2380-2389.

Developing an engineering design for a system that is highly unlikely to produce cost-competitive hydrogen does not in my mind lend itself to publication in a high impact journal. Rather it belongs in an engineering journal.

In summary, for these new engineering designs, I'm inclined to request a technoeconomic analysis on them to see what the possibilities are for producing cost-competitive hydrogen. It is more that an engineering design these days, the metric is the final cost of hydrogen and tools are available to put forth a projected cost based on the specific design.

Publication in Nature Communication is not recommended.

We understand the argument of the Reviewer #1 concerning the metric of a viable commercial product. However, it is not our intention to present our concept as the structure of choice for a minimal projected cost of hydrogen in a commercial product. The reviewer is right that for such an assessment a complete technoeconomical analysis is required with the tools available from the attached references.

However, this is out of the scope of this work and will require modifications to the models. Future works with such modifications could help to create an in depth technoeconomical analysis of the reported approach. For now, we think that it is beneficial to first proof that the concept is working physically before assessing its economical viability. A similar approach was taken for the 'louvered cell' design which was first introduced by Walczak et al. (ChemSusChem 8, 544 (2015)), subsequently evaluated with regard to its energy balance by Sathre et al. (Energy Environ. Sci., 2016, 9, 803, DOI:

10.1039/c5ee03040d), and finally assessed in the work by Shaner et al. (Energy & Env. Sci, 2016, DOI: 10.1039/c5ee02573g) for its cost competitiveness.

One conclusion is that with the STH efficiencies projected for solar hydrogen generation no sustainable technology is cost competitive to the generation of hydrogen from steam reforming. However, this figure may change if new materials are explored which increase the efficiency of the device. In this regard we would like to point out that the presented device design is not restricted to thin-film silicon but can be applied to any thin-film PV technology that can be deposited on large areas.

To address the concern of the reviewer and point out the limitations of the study we have revised the discussion of our study and point out possible future works on this topic together with the references as an entry point.

Reviewer #3 (Remarks to the Author):

I have now read the new version of the manuscript by Turan. et al. My general impression is fairly unchanged from the last time I saw it. They do construct working devices, the discussion is sound, the presentation is decent and the topic of upscaling and the associated problems are important for the application of water splitting systems and is something that deserves more attention. With the understanding that it is more of an engineering paper I think this manuscript deserves to be published. I think that the changes they have made in response to the reviews comments are reasonable. Think that the comment from rewire 1 concerning the separation of the reaction gases are a bit unfair as incorporating membranes that can do this is a rather straight forward geometrical exercise given the suggested experimental setup.

I have no additional comments and are in favour of acceptance for publication.

We thank the reviewer for the positive response and feedback on our work.